# Preventability of Colorectal Cancer in Saudi Arabia: Fraction of Cases Attributable to Modifiable Risk Factors in 2015–2040

**DOI:** 10.3390/ijerph17010320

**Published:** 2020-01-02

**Authors:** Abdulmohsen Al-Zalabani

**Affiliations:** Department of Family and Community Medicine, College of Medicine, Taibah University, Madinah 41541, Saudi Arabia; aalzalabani@gmail.com; Tel.: +966-505-654549

**Keywords:** colorectal neoplasms, population attributable fraction, lifestyle, exercise, tobacco smoking, risk factors, primary prevention, obesity, overweight

## Abstract

A rise in colorectal cancer (CRC) burden is expected around the globe. This study aimed to determine the population attributable fractions (PAFs) of CRC cases contributed by modifiable risk factors in Saudi Arabia. The PAF was calculated for modifiable risk factors with strong evidences of a causal association with CRC. CRC incidence was obtained from the National Cancer Registry, relative risks were retrieved from recent meta-analysis studies, and the prevalence of exposure to risk factors was obtained from national surveys. Conventional statistical formulas were used to calculate PAFs from registered CRC cases, stratified by sex. Three scenarios were proposed to make projections and present the expected effects of prevention interventions on the number of CRC cases in Saudi Arabia for 2025–2040. The results showed the largest fraction of attributable CRC cases among men and women was contributed by physical inactivity (16.13% and 16.45%), followed by excess weight (obesity: 9.71% and 6.93%; overweight: 6.05% and 1.9%); and tobacco smoking (current smoker: 3.04% and 0.18%; former smoker: 3.29% and 0.12%). We estimated that the number of projected cases attributable to physical inactivity, smoking, and excess weight in men and women would increase from 807 and 315 in 2025 to 1360 and 556 in 2040, respectively. In conclusion, physical inactivity, being overweight or obese, and tobacco smoking are major lifestyle factors affecting the incidence of CRC in Saudi Arabia. Prevention interventions and public health programs to reduce their prevalence are warranted.

## 1. Introduction

Colorectal cancer (CRC) constitutes a high disease burden globally, and it has been projected to further increase by 60% by the year 2030 [1]. The incidence and mortality rates of CRC can vary by up to 10-fold among different countries. The highest rates are observed in developed countries, where they maintain a stable trend. However, rates increase rapidly in developing countries [1], which may be attributed to the adoption of Western lifestyles by their populations.

Lifestyle and other modifiable risk factors play an important role in the development of CRC. Evidence shows that the risk of CRC increases with obesity, tobacco use, consumption of processed and red meat, and alcohol ingestion, whereas it decreases with physical activity and consumption of whole grains, dietary fiber, and dairy products [2]. Therefore, these modifiable risk factors can be potentially modified to prevent CRC.

In Saudi Arabia, CRC is the most common type of cancer among men and the third most common type among women [3,4]. Notably, most of the reported CRC cases in Saudi Arabia are diagnosed during clinical evaluations rather than through screening programs. A national survey showed a low utilization of CRC screening programs in Saudi Arabia, with only 5.64% of the Saudi elderly population reporting having undergone CRC screenings [5]. This could be related to various factors including: low awareness of their existence among the population [6], low physician adherence to recommend CRC screening [7], and the lack of a national policy [8]. However, it should be noted that a successful adoption of the screening recommendations would have an impact on the reported number of CRC cases [9].

The age-standardized rate (ASR) of CRC in Saudi Arabia increased from 5.0 per 100,000 persons in 2001 to 9.6 per 100,000 persons in 2015 [10]. This increase is related not only to an increase in the proportion of the population represented by the elderly [10], but also to the increased prevalence of various lifestyle-related risk factors [11,12]. Understanding the contribution of each risk factor in the burden of CRC could help the decision-making process and guide resource allocation for prevention interventions.

This study aimed to determine the population attributable fraction (PAF) contributed by modifiable risk factors in the incidence of CRC in Saudi Arabia and to create projections on the impact of prevention interventions on the attributable cases of CRC.

## 2. Materials and Methods

The PAF is a measure that indicates how much the incidence of a certain disease would be reduced if the population were unexposed to a specific risk factor, assuming a causal association [13]. The PAF calculation requires information on the estimates of CRC incidence, prevalence of risk factors in the target population, and the relative risk (RR) of each factor in the incidence of CRC. The following sections describe the selection of risk factors and the sources of data on each estimate.

### 2.1. Selection of Modifiable Risk Factors

Risk factors classified by the World Cancer Research Fund (WCRF) [2] as having ‘convincing evidence’ or by the International Agency for Research on Cancer (IARC) [14] as having ‘sufficient evidence’ of a causal association with CRC were selected. These included physical inactivity, overweight or obesity, and tobacco use. Other risk factors such as consumption of processed meat and alcoholic drinks were not included in the study because no suitable exposure prevalence data for their consumption in Saudi Arabia were available.

### 2.2. CRC Incidence

The incidence rate of CRC was obtained from the most recent Cancer Incidence Report [15] issued by the National Cancer Registry, which provided information for the year 2015. The National Cancer Registry is a nationwide registry of cancer patients. The details of data collection, management, and analysis for the Cancer Incidence Report are reported elsewhere [15].

To estimate the changes in ASR trends, ASR data from 1994 to 2015 were retrieved from the Cancer Incidence Reports [15,16] and entered into the Joinpoint Regression Program version 4.7.0.0 (National Cancer Institute, Bethesda, MD, USA) [17]. The regression model and statistical tests used by the Joinpoint Regression program are described elsewhere [18].

### 2.3. Prevalence of Modifiable Risk Factors

The lag period between exposure to risk factors and development of CRC was assumed to be 10 to 15 years [19]. Thus, information on exposure prevalence was obtained from a national survey [11] conducted by the Saudi Ministry of Health in collaboration with the World Health Organization (WHO) in 2004. The methodology of the survey is detailed elsewhere [11]. This survey provided information on the prevalence of tobacco use, overweight, obesity, and physical inactivity [11,20]. Data on the prevalence of physical inactivity were further obtained from a more detailed publication [20] based on the same survey dataset.

### 2.4. Relative Risk

The relative risk (RR) estimates for various risk factors related to the incidence of CRC were derived from recent systematic reviews and meta-analyses [21,22,23] that have also been used in similar studies [19,24]. Separate estimates for men and women were used if available; otherwise, the overall RR was used. Global meta-analyses were used because there is no meta-analysis conducted based on local data.

### 2.5. Calculation of PAF

PAF was calculated based on estimates of prevalence of each risk factor in Saudi Arabia and on RR estimates identified in well-designed systematic reviews. The following formula was used for calculating PAF:PAF = ((RR − 1) × P_e_)/(1 + (RR − 1) × P_e_)(1)
where RR is the relative risk for a specific risk factor, and P_e_ is the proportion of the population exposed to that risk factor.

The combined PAF for all factors was calculated using the following formula:(2)$$PAF_{combined}=1-\;\prod_{n=1}^{total}(1-\left(PAF_{n}\right))$$
where *PAF_combined_* represents the aggregate PAF for all factors included in the analysis, and *PAF_n_* represents the specific PAF for each risk factor [25]. This formula avoids overestimation of the combined PAFs for all risk factors.

### 2.6. Projections of Attributable CRC Cases in Saudi Arabia

The projected number of CRC cases from 2025 to 2040 was obtained using the IARC online interactive query tool ‘Cancer Tomorrow’ [26]. ‘Cancer Tomorrow’ provides projected estimates for CRC incidence in each country based on projected changes in the country’s demography and the ASR for CRC. In the current study, the query tool was set to assume that there would be no changes in the ASR for CRC in Saudi Arabia during the estimation period, to obtain conservative estimates for the projected number of cases. Since the prevalence estimates used in the current study provided information about Saudi citizens, whereas the IARC tool provided estimates for all cases in the country, the projected estimates for Saudi citizens were calculated based on the average proportion of Saudi patients in the overall population of CRC patients reported in the Cancer Incidence Reports for 2014–2015 [15,16]. To explore the potential effect of prevention interventions, three scenarios were tested to calculate the impact of the lifestyle risk factors on the projected cases of CRC for 2025–2040. The first scenario assumed that the prevalence of all risk factors would remain constant. The second and third scenarios assumed that the prevalence of each risk factor would be reduced every five years by 5% and 10%, respectively. In each scenario, a lag period of 10 years from exposure to the risk factor to development of CRC was used.

## 3. Results

### 3.1. CRC Incidence

In 2015, CRC accounted for 12.2% of newly diagnosed cancer cases [15]. Particularly, 810 men and 655 women were newly diagnosed with CRC in that year. The ASR was 11.2 per 100,000 men and 9.1 per 100,000 women. Figure 1 and Figure 2 show the trend of the ASR for CRC among men and women from 1994 to 2015 based on data from the Cancer Incidence Report. The average annual percentage change was 4.9% (95% confidence interval (CI): 3.0–6.7) among men and 3.7% (95% CI: 2.0–5.5) among women.

### 3.2. Prevalence of Risk Factors

Table 1 presents the prevalence rates of the selected risk factors for men and women. Compared to other selected risk factors, physical inactivity had the highest prevalence rate in both men and women (60.1% and 72.9%, respectively) [20]. The prevalence of obesity was higher among women, while the prevalence of overweight was higher among men. The smoking rate was higher in men than in women; the prevalence of current smokers was 20.9% for men and 1.2% for women [11].

### 3.3. Relative Risk Data

As shown in Table 1, physical activity decreases the risk of CRC, with a RR of 0.76 (95% CI: 0.71–0.82) among men and 0.79 (95% CI: 0.71–0.88) among women [22]. On the contrary, excess weight is associated with an increased risk of CRC in both men and women. A recent systematic review [23] reported a RR of 1.38 (95% CI: 1.32–1.44) and 1.17 (95% CI: 1.06–1.30) among obese men and women, respectively. The same systematic review reported a RR of 1.17 (95% CI: 1.12–1.22) and 1.07 (95% CI: 1.01–1.14) among overweight men and women, respectively. Tobacco use was also associated with an increased risk of CRC, and a systematic review reported that current smokers have a RR of 1.15 (95% CI: 1.00–1.32), while former smokers have a RR of 1.20 (95% CI: 1.04–1.38) for CRC [21].

### 3.4. PAF

Table 2 presents the PAF and number of CRC cases attributable to each risk factor. Physical inactivity was linked to the largest fraction of attributable CRC cases in Saudi Arabia, with a similar PAF among men (16.13%) and women (16.45%). In 2015, approximately 130 male and 108 female CRC cases in Saudi Arabia were attributable to physical inactivity. Further, excess weight contributed to higher PAF estimates in men than in women. In total, 9.71% of male and 6.93% of female CRC cases were attributed to obesity. Further, 6.05% of male and 1.9% of female CRC cases were attributed to overweight, corresponding to a total of 124 CRC cases attributable to obesity and 61 CRC cases attributable to overweight in Saudi Arabia in 2015.

The proportion of CRC cases caused by current and former tobacco smoking in men in Saudi Arabia was 3.04% and 3.29%, respectively. The number of cases attributable to smoking was lower among women than among men (0.18% for current smokers and 0.12% for former smokers).

### 3.5. Projections

The projected number of CRC cases in Saudi Arabia for 2025–2040 is presented in Table 3 and Figure 3. In the basic scenario of a constant prevalence of the risk factors (i.e., similar to the values presented in the current study), the estimated number of projected cases attributable to physical inactivity, smoking, and excess weight would increase from 807 male cases and 315 female cases by 2025, to 1360 male cases and 556 female cases by 2040. By contrast, in the scenario of a 5% relative reduction in the prevalence of each risk factor every five years, the number of attributable male and female cases by 2040 would be 1206 and 489, respectively. Moreover, a 10% reduction in the prevalence of each risk factor every five years would lead to a lower increase in the number of attributable cases, expecting 1058 male cases and 427 female cases by 2040.

## 4. Discussion

The current study analyzed the trend of the ASR for CRC in Saudi Arabia from 1994 to 2015 and estimated the PAF for modifiable risk factors. The findings demonstrated that physical inactivity, overweight and obesity, and tobacco smoking were linked to one-third of male and one-fourth of female CRC cases in Saudi Arabia in 2015. Further, the analysis revealed the estimated number of CRC cases that can be prevented by 2025–2040 if the exposure to these risk factors was reduced.

The ASR for CRC showed an overall increase over the past years. However, the rate of increase in recent years is lower than that of earlier years, when the registry started. For example, the annual percent change in the CRC rate among men during the period between 2005 and 2015 was 2.15%, whereas that one from 1997 to 2005 was 12.85%, indicating that the incidence of CRC may be reaching a plateau. This result may be related to the establishment of the registry, which rapidly improved CRC reporting in the initial phases.

The PAF contributed by physical inactivity was the largest for both men and women, a result that is related to the high prevalence of physical inactivity [20] in the general population. The PAF contributed by smoking was the smallest, which is related to the fact that smoking is the risk factor with the lowest prevalence when compared to physical inactivity and body fatness. In fact, the smoking-associated PAF is smaller among women than among men, which can also be attributed to the lower prevalence of smoking in the female population. However, it should be noted that these figures are bound to change in the future as the smoking rate in women is increasing at a much higher rate than in men. From 2004 to 2013, the smoking prevalence among women in Saudi Arabia had an increase of 57.1% (from 0.7% to 1.1%), whereas among men it had an increase of 40.5% (from 15.3% to 21.5%). This pattern could be explained by the cultural changes in Saudi society, which has progressively accepted smoking habits among women.

The findings of the current study show the significance of physical inactivity, obesity, and tobacco smoking in the burden of CRC in Saudi Arabia and emphasizes the potential of their modification on CRC prevention. The Saudi Arabian government has already recognized the contribution of changing lifestyles in the burden of diseases in Saudi Arabia and has thus developed future health plans that focus on public health measures and prevention initiatives. The Vision 2030 and the National Transformation Program 2020 (NTP) target lifestyle risk factors among the strategic objectives of the health system [27,28]. For example, the ‘Vibrant Society with Fulfilling Lives’ theme of Vision 2030 laid the following objective: ‘to increase the ratio of individuals exercising at least once a week from 13% to 40%’. This objective was translated into various multidisciplinary initiatives in the ‘Quality of Life Program 2020’ and other programs [29]. In the NTP, the Ministry of Health set a strategic objective to ‘improve public health services with focus on obesity and smoking’ aiming to reduce the prevalence of smoking by 2% and the prevalence of obesity by 1% from the baseline by 2020 [27]. The current refocus of healthcare efforts on disease prevention and public health interventions was a long-awaited shift necessitated by the growing prevalence of lifestyle risk factors including physical inactivity, obesity, and tobacco use [11,12]. Prevention interventions targeting lifestyle risk factors, in addition to other interventions like screening programs, can alleviate the risk of CRC in Saudi Arabia. However, a challenge that needs to be addressed is the absence of a surveillance system for behavioral risk factors in Saudi Arabia. This lack of longitudinal data precludes the analysis of possible trends and the evaluation of the interventions aimed at reducing the prevalence of behavioral risk factors.

Comparison of PAF estimates between the current analysis and similar studies in other countries shows a wide range of variation [19,24,30]. In Malaysia for example, physical inactivity contributed to 10.1% and 11.6% of the CRC cases among men and women, respectively; while overweight contributed to 4.5% and 0.9%, respectively [30]. In New Zealand, physical inactivity, obesity, and smoking contributed to 4.4%, 9.0%, and 2.5% of all CRC cases, respectively. These PAF variations are expected because of intrinsic differences in the prevalence of risk factors included in the PAF calculation among populations.

The strengths of the current study include the use of nation-representative data of exposure prevalence, the use of CRC incidence data from the National Registry, and the use of recent systematic reviews and meta-analyses to obtain the RR estimates. The study provided a national estimate for PAFs contributed by modifiable risk factors with evidence of a causal relationship to CRC incidence in Saudi Arabia. These data will be useful for policy makers in healthcare planning and resource allocation.

However, this study also has limitations, including those related to the data source and nature of PAF estimates. First, prevalence data were based on self-reports, which tend to underestimate the prevalence of behavioral risk factors and thus will lead to underestimation of the calculated PAFs. Second, the selection of factors was confined to those that have been classified as having convincing/sufficient evidence, and this could also lead to underestimation of PAFs and, consequently, to underestimation of the prevention potential of modifiable risk factors. The WCRF Report also identifies other modifiable risk factors with ‘probable’ evidence of causal association to CRC, including the consumption of whole grains, dietary fiber, dairy products, and red meat [2]. Including these factors in analyses would result in higher PAF estimates for both sexes. Third, a national estimate for the prevalence of consumption of processed meat and alcohol was not available and thus could not be included in the analyses. Including these two factors would increase PAF estimates. The current study emphasizes the importance of having national estimates of various modifiable risk factors through regular surveys using standardized methodologies that facilitate comparison and trend analyses. Finally, having multiple components in the calculation of PAF precluded the provision of the traditional confidence intervals [19].

## 5. Conclusions

In conclusion, physical inactivity, overweight and obesity, and tobacco smoking are major contributing factors to the incidence of CRC in Saudi Arabia. Further, the findings underscore that reductions on the exposure to these risk factors may have great potential for preventing CRC. Therefore, healthcare strategies in Saudi Arabia should focus on enhancing prevention interventions and public health programs to reverse the increasing prevalence of modifiable risk factors.

## Figures and Tables

**Figure 1 ijerph-17-00320-f001:**
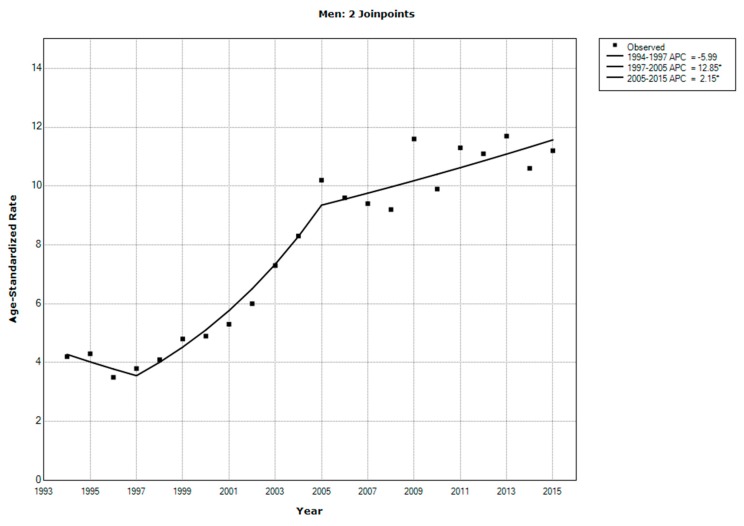
Trends in the age-standardized incidence rates for colorectal cancer among men in Saudi Arabia, 1994–2015. * indicates that the annual percent change is significantly different from zero at the alpha = 0.05 level. Final selected model: 2 Joinpoints.

**Figure 2 ijerph-17-00320-f002:**
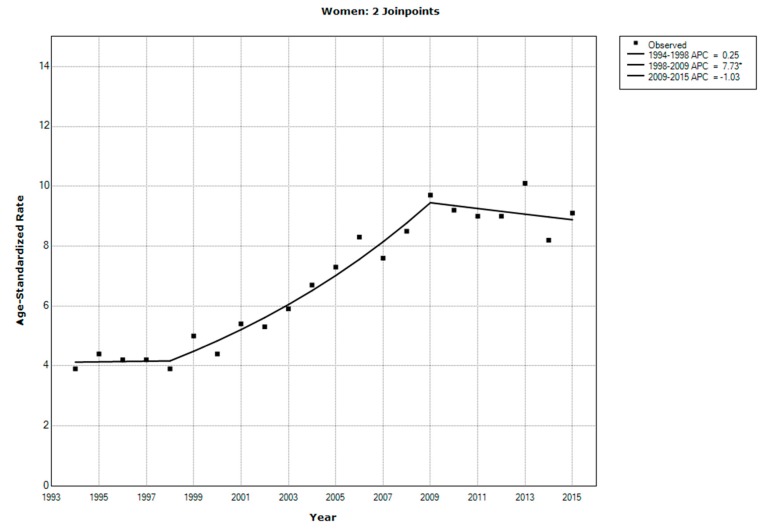
Trends in the age-standardized incidence rates for colorectal cancer among women in Saudi Arabia, 1994–2015. * indicates that the annual percent change is significantly different from zero at the alpha = 0.05 level. Final selected model: 2 Joinpoints.

**Figure 3 ijerph-17-00320-f003:**
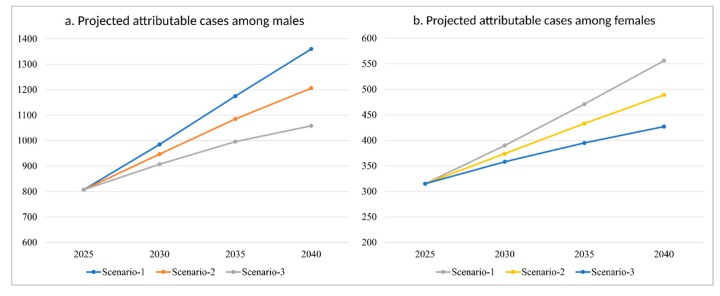
Projections of combined attributable CRC cases under different scenarios for (**a**) males and (**b**) females in Saudi Arabia, 2025–2040.

**Table 1 ijerph-17-00320-t001:** Prevalence and relative risk estimates for the selected risk factors.

	Prevalence (in Year 2004)	Relative Risk	Source of Relative Risk
**Men**			
Physical inactivity	60.1%	0.76 (0.71, 0.82)	[22]
Obesity	28.3%	1.38 (1.32–1.44)	[23]
Overweight	37.9%	1.17 (1.12–1.22)	[23]
Current smokers	20.9%	1.15 (1.00–1.32)	[21]
Former smokers	17.0%	1.20 (1.04–1.38)	[21]
**Women**			
Physical inactivity	72.9%	0.79 (0.71, 0.88)	[22]
Obesity	43.8%	1.17 (1.06–1.30)	[23]
Overweight	27.6%	1.07 (1.01–1.14)	[23]
Current smokers	1.2%	1.15 (1.00–1.32)	[21]
Former smokers	0.6%	1.20 (1.04–1.38)	[21]

**Table 2 ijerph-17-00320-t002:** Population attributable fractions (PAFs) and number of colorectal cancer (CRC) cases attributable to each risk factor in Saudi Arabia, 2015.

	PAF (%)		Attributable Cases (2015)	
	Men	Women	Men	Women
Physical inactivity	16.13	16.45	130	108
Obesity	9.71	6.93	79	45
Overweight	6.05	1.9	49	12
Current smokers	3.04	0.18	25	1
Former smokers	3.29	0.12	27	1
All factors	33.29	23.94	310	167

**Table 3 ijerph-17-00320-t003:** Projections of combined population attributable fractions and attributable CRC cases under different scenarios, 2025–2040.

Year	Projected Cases	Scenario-1		Scenario-2		Scenario-3	
		PAF ^1^	Attrib. Cases ^1^	PAF	Attrib. Cases	PAF	Attrib. Cases
**Men**							
2025	2423	33.29%	807	33.29%	807	33.29%	807
2030	2958	33.29%	985	32.00%	947	30.67%	907
2035	3529	33.29%	1175	30.74%	1085	28.21%	996
2040	4085	33.29%	1360	29.52%	1206	25.89%	1058
**Women**							
2025	1314	23.94%	315	23.94%	315	23.94%	315
2030	1628	23.94%	390	22.96%	374	21.96%	358
2035	1967	23.94%	471	22.01%	433	20.10%	395
2040	2321	23.94%	556	21.09%	489	18.38%	427

^1^ PAF: population attributable fraction; Attrib. Cases: attributable cases.

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
