# Peer review of "Preventability of Colorectal Cancer in Saudi Arabia: Fraction of Cases Attributable to Modifiable Risk Factors in 2015–2040"

_ijerph, 2020, doi:10.3390/ijerph17010320_

Round 1

Reviewer 1 Report

Abdulmohsen Al-Zalabani analyzed the CRC statistics to attribute physical inactivity, overweight or obese and also tobacco smoking to be the major lifestyle factors affecting the incidence of CRC in Saudi Arabia.

Overall, the manuscript described a lot of evidence for CRC status in Saudi Arabia and made a clear conclusion about modifiable risk factors based on the determination of PAF of CRC cases. Below, I have two specific comments/questions:

Table 1, the author used different references to rank the prevalence of several risk factors, that was not convincible. And the author did not address why the data in 2004 was used, which should be replaced with more updated/latest data.

The data in Table 3 could be better elucidated by plotting that in to 1 or 2 figures.

Author Response

Point 1: Table 1, the author used different references to rank the prevalence of several risk factors, that was not convincible. And the author did not address why the data in 2004 was used, which should be replaced with more updated/latest data.

Response 1: Thank you. In Table 1, all prevalence estimates were derived from the same study (STEPs survey conducted in year 2004). Relative risk estimates were derived from different meta-analyses according to the risk factor (there was no meta-analysis that provided relative risk estimates for all investigated risk factors).

Regarding the selection of 2004 data rather than more recent data, the year 2004 data was intentionally used to account for the lag period happening between exposure to the risk factors and the development of cancer. This justification is now highlighted in section 2.3 line 81.

Point 2: The data in Table 3 could be better elucidated by plotting that in to 1 or 2 figures.

Response 2: Thank you for the suggestion. As suggested, Figure 3 is now added to plot the projections of attributable cases in males and females.

Reviewer 2 Report

The author provides interesting data to understand colon cancer cases and trends in Saudi Arabia.  Obesity, smoking and lack of physical activity were found to be contributing factors.

How does other disease affect this?  No contribution from cardiovascular disease, diabetes has been mentioned. Website for the Saudi Arabia cancer registry should be included. What are the risk/hazard ratios for the PAFs for Saudi populations? What would be the recommendation to alleviate the risk? Will ASR benefit from Colonoscopy or FOBT?

Author Response

Point 1: How does other disease affect this? No contribution from cardiovascular disease, diabetes has been mentioned.

Response 1: Thank you. The current study focused on three modifiable risk factors amenable to change by preventive interventions. Although other factors contribute to the burden of disease, PAF calculation does not depend on the contribution of other factors which allow us to focus on the three factors. Discussion of the contribution of other risk factors is beyond the scope of the current study.

Point 2: Website for the Saudi Arabia cancer registry should be included.

Response 2: Done. The links to the website of the NCR reports are now added in Ref 15 and 16.

Point 3: What are the risk/hazard ratios for the PAFs for Saudi populations?

Response 3: Thank you. I agree that using estimates from local population is preferred. However, I used the risk/hazard ratios from recent global systematic reviews and meta-analyses because there are no studies in Saudi Arabia reporting these estimates for various risk factors. Moreover, the use of international estimates makes the results comparable with similar international studies using the same methodology. A statement was added to section 2.4 to elaborate this point.

Point 4:

What would be the recommendation to alleviate the risk? Will ASR benefit from Colonoscopy or FOBT?

Response 4: Thank you. Although this study emphasized the contribution of modifiable risk factors to the incidence of CRC and highlights the potential for prevention, it is not possible to recommend specific interventions based on the current study. Discussion paragraph in lines 202 to 221 focused on the general recommendations for refocusing healthcare system to address these risk factors and consequently reduces the risk of CRC. Also, the current study cannot determine the effectiveness of screening programs. I added a statement in lines 216-218 to emphasize that other interventions can also play a role in CRC prevention.

Round 2

Reviewer 1 Report

The author addressed most of the prior concerns indicated in the first review satisfactorily. The revised manuscript is better organized than the previous version and now acceptable for publication.